# Health determinants among refugees in Austria and Germany: A propensity-matched comparative study for Syrian, Afghan, and Iraqi refugees

Daniela Georges[1]*, Isabella Buber-Ennser[2], Bernhard Rengs[2], Judith Kohlenberger[3], Gabriele Doblhammer[1,4]

1 Department of Sociology and Demography, University of Rostock, Rostock, Germany, 2 Vienna Institute of Demography (OeAW), Wittgenstein Centre for Demography and Global Human Capital (IIASA, OeAW, University of Vienna), Vienna, Austria, 3 Institute for Social Policy, Vienna University of Economics and Business, Vienna, Austria, 4 German Center for Neurodegenerative Diseases, Bonn, Germany

☉ These authors contributed equally to this work.
* daniela.georges@uni-rostock.de

**Data Availability Statement:** The data underlying this study are owned by third party sources and can be accessed following the the information in

## Abstract

In recent years, Germany and Austria have been among the leading European receiving countries for asylum seekers and refugees (AS&R). The two countries have cultural and economic similarities, but differ, for example, in their health care systems, with AS&R having unrestricted access to health services upon arrival in Austria, but not in Germany. This study investigates the determinants of health among refugees in Austria and Germany, and how these determinants differ between the two countries. We analyze comparable and harmonized survey data from both countries for Syrian, Afghan, and Iraqi nationals aged 18 to 59 years who had immigrated between 2013 and 2016 (Germany: n = 2,854; Austria: n = 374). The study adopts a cross-sectional design, and uses propensity score matching to examine comparable AS&R in the two receiving countries. The results reveal that the AS&R in Germany (72%) were significantly less likely to report being in (very) good health than their peers in Austria (89%). Age and education had large impacts on health, whereas the effects of length of stay and length of asylum process were smaller. Compositional differences in terms of age, sex, nationality, education, and partnership situation explained the country differences only in part. After applying propensity score matching to adjust for structural differences and to assess non-confounded country effects, the probability of reporting (very) good health was still 12 percentage points lower in Germany than in Austria. We conclude that many of the determinants of health among AS&R correspond to those in the non-migrant population, and thus call for the implementation of similar health policies. The health disadvantage found among the AS&R in Germany suggests that removing their initially restricted access to health care may improve their health.

the Materials and methods section. The German Socio-Economic Panel (GSOEP) data contain potentially sensitive information and due to legal restrictions by the German data protection law, GSOEP data from this study are only available upon request. The scientific use file of the German Socio-Economic Panel (GSOEP) is made available for scientific research by the German Institute for Economic Research (DIW) at doi: 10.5684/soep. iab-bamf-soep-mig.2016. The use of anonymized GSOEP data is subject to strict standards in the data provision and are reserved exclusively for research use. GSOEP data are available free of charge as scientific use files after requesting a data distribution contract. The form is available online: https://www.diw.de/documents/ dokumentenarchiv/17/diw_01.c.88926.de/soep_ application_contract.pdf. For further information the GSOEP hotline at either soepmail@diw.de or +49 30 89789-292 can be contacted. The ReHIS data are made available upon registration for scientific research by the Austrian Social Science Data Archive (AUSSDA) at doi:10.11587/7LX1BD. The anonymous IDs of the respondents selected for the current study are provided upon request.

**Funding:** This work was supported by the Austrian Federal Ministry of Education, Science and Research; the Austrian Federal Ministry of Labour, Social Affairs, Health and Consumer Protection; the Fonds Soziales Wien (FSW); Common Health Goals of the "Rahmen-Pharmavertrag", a cooperation between the Austrian pharmaceutical industry and the Austrian social insurance [grant number 99901007700; initials of author who received the award: JK]. The funders had no role in the study design, data collection and analysis, decision to publish, or preparation of the manuscript.

**Competing interests:** The authors have declared that no competing interests exist.

## Introduction

In recent years, Europe has been the destination of large inflows of refuge-seeking individuals, with more than 4.6 million individuals arriving in the EU-28 countries over a five-year period [1]. Large shares of these asylum seekers came from Syria, Afghanistan, and Iraq. To date, the political, societal, and scientific discourses on this wave of refugees have focused mainly on its effects on the economies and welfare systems of the receiving countries in Europe [2–5], while less attention has been paid to refugees' health and their access to health services [6–13]. While a number of studies have examined the mental and physical health of refugees in large refugee camps and in low-income countries [14–17], the health of refugees in high- or medium-income country contexts remains under-researched [9–11, 13]. The previous studies that have examined this topic have found that compared to the health status of the total population, AS&R in Germany have better physical but worse mental health [18], while male AS&R in Austria have better self-rated health [19].

This research gap has important consequences, as health is an individual's most important resource for successful integration into a society and the labor market of the receiving country [20, 21]. There are numerous determinants of refugees' health. Among AS&R, being female [7, 22] and being older [7, 12] are associated with worse health, while having higher levels of education [23] and (family) social support [22, 24] are associated with better health. Moreover, the health of AS&R varies by their country of origin [6, 22]. Refugee-specific determinants of health include the circumstances and experiences of individuals before they fled, during their journey, and after their arrival in a host country [16, 20, 21]. Moreover, access to health care services in the destination country has been shown to be a key factor in the health of AS&R. There is, for example, evidence that when AS&R face no formal access barriers to care, they tend to be in better health and have higher levels of social inclusion. Moreover, the lack of such barriers might reduce public health expenditures [8, 25–27].

Germany and Austria are two high-income European countries with high gross domestic product (GDP) levels and above-average medical care standards [28, 29]. Both countries have received large numbers of asylum seekers. During the last five years, about 1.8 million asylum applications have been filed in Germany and 197,000 applications have been filed in Austria [1]. In this period, the number of individuals who were officially granted asylum (including subsidiary protection and protection on humanitarian grounds) was roughly 1.1 million in Germany and 109,000 in Austria [30, 31].

In both countries, health care expenditures are equivalent to 10–11% of the GDP, a share that is above the EU-28 average (numbers refer to 2016; EU-28 includes the 28 member states of the European Union as of 2016) [29]. Health insurance is granted to both asylum seekers and refugees, but in different ways. In Austria, as legally mandated in the Austrian General Social Insurance Law from 2004, individuals can make use of all health care services provided by the medical insurance system upon submission of their asylum application. This includes access to public hospitals, psychological treatments, and medications. Therefore, asylum applicants have the same formal access to the health care system as the resident population [11]. In contrast, as regulated by the German Social Welfare Law for asylum seekers from 1993, Germany provides limited access to asylum seekers up to 15 months after they have submitted their asylum application, including essential medical treatment, vaccinations, and pregnancy care. After that period, asylum applicants receive regular medical care and have the same access to health care as the German resident population. Moreover, once applicants have received a positive decision on their application, refugees enjoy unlimited access to health care.

Taking the cultural and economic similarities as well as the differences in health policy in the two countries into account, our paper has two central research aims: first, we want to provide insight into self-rated health (SRH) and determinants of health among AS&R in Germany and Austria; second, we want to put the factors that may contribute to the differences between the two countries in perspective. We hypothesize that we will find 1) differences in health outcomes by sociodemographic characteristics among AS&R, and 2) health differences among AS&R in Germany and Austria.

## Materials and methods

### Data and population

This study uses data from the IAB-BAMF-SOEP-Refugee Survey 2016 (for Germany) and from the Refugee Health and Integration Survey (ReHIS) (for Austria). The IAB-BAMF-SOEP-Refugee Survey 2016 [32] includes responses from AS&R who arrived in Germany between 2013 and 2016. Interviews were conducted as CAPIs (computer-assisted personal interviews) in Arabic, Kurmanji, Farsi/Dari, Urdu, German, and English. The translation was carried out conscientiously by two translators, and the responses were subsequently harmonized [33]. The random sample was based on the German Central Register of Foreign Nationals (which contains information about all foreign nationals in Germany), and included 4,527 individuals aged 18 years or older [33]. The response rate was roughly 50%, with only a small proportion of nonresponse being refusal (~10%) or due to illness or nursing care (<1%) [34]. The questionnaire consisted of a detailed personal questionnaire and a household questionnaire. The survey collected information on the health, migration, educational, and employment biographies of AS&R, as well as on their reasons for fleeing, the routes they took, and their personality and attitudes [35, 36].

The ReHIS was conceptualized as an interim survey within FIMAS, a project on the labor market participation of Syrian, Afghan, and Iraqi refugees in Austria. The ReHIS survey was an interim survey between the second and third wave of the FIMAS+INTEGRATION panel [37]. The FIMAS+INTEGRATION sample contained 780 persons that agreed to participate in the interim survey and provided contact details. The response rate was 68%, where the majority could not be reached due to incorrect contact details or not picking up the phone after multiple contact attempts. Only 6% of the non-responding persons refused to participate. The interviews were carried out in early 2018 as CATIs (computer-assisted telephone interviews) mainly in Arabic and Farsi/Dari as well as few interviews in German. The ReHIS was based on selected EHIS (European Health Interview Survey) items and the IAB-BAMF-SOEP-Refugee Survey 2016. The sample consisted of 515 persons aged 18–61 years who arrived in Austria between 2011 and 2018 [38]. The questionnaire spanned 50 items focused on psychosocial and physical health, barriers and patterns of health care utilization, and individual characteristics [38, 39].

For further information on the field phase and data collection, we refer to two other studies [33, 40].

For reasons of comparability, the current study is restricted to AS&R who are Syrian, Afghan, or Iraqi nationals aged 18–59 years who immigrated between 2013 and 2016. These three nationalities have made up a large share of the asylum seekers in Europe in recent years, especially in Austria and Germany [1]. Our sample comprises 2,854 respondents in Germany and 374 in Austria.

### Health measure and control variables

Our German data source (IAB-BAMF-SOEF-Refugee Survey 2016) had a broad focus on refugees' lives in their host country of Germany. Each respondent was asked questions about his/

her living situation, legal status, vocational training, language skills, employment, state bene-fits, religion, worries and concerns, political attitudes and interests, attitudes toward women, family situation, and general life satisfaction. A short cognitive test (the "symbols and numbers test" to test the speed of perception and fluid intelligence) was also administered. In addition, the respondents were asked several questions related to their health, including about their general health, as well as their physical and mental health. The questionnaire was validated via qualitative pretests [34]. During the interview, the interviewer and the respondent were able to simultaneously look at the questionnaire in German and in the respondent's language in order to minimize language barriers [41].

The focus of our Austrian data source (ReHIS) was on psychological health and access to medical care and integration services among refugees in Austria. The respondents were asked questions about their general health, physical health, psychological well-being, experiences with the Austrian healthcare system (including unmet needs), concerns and worries, and the extent to which they feel welcome in the host country. Information on each respondent's demographic characteristics, education, employment, legal status, language proficiency, and family context was derived from an interview conducted shortly before the ReHIS in the framework of the embedded survey on the labor market participation of refugees in Austria FIMAS [37]. All of the relevant question blocks and items were directly taken from the well-established EHIS survey, and from the IAB-BAMF-SOEP-Refugee Survey 2016. After rigorous technical and internal tests/mockup interviews were conducted with native speakers, an intensive pretest phase with 20 completed interviews in Arabic and Farsi was undertaken with refugees residing in Austria [37].

In our comparative study, the main variable of interest is SRH, which was included in both questionnaires. The exact wording of the question was "How would you describe your current state of health? (1) Very well, (2) well, (3) satisfactory, (4) Not very good, (5) poor" in the IAB-BAMF-SOEF-Refugee Survey; and was "In general, would you say your health is (1) very good, (2) good, (3) acceptable, 4 (fair), (5) bad" in the ReHIS. Answer options were dichotomized into "(very) good" (comprising the answers (1) and (2)) and "less than good" (including answer options (3), (4), and (5)). The set of control variables refers to the WHO's "Social Determinants of Health" framework, which describes the interrelation of structural determinants of health inequities (the socioeconomic and political context in the destination country, and the socioeconomic position in the country of origin), intermediary determinants of health (material circumstances, biological and behavioral factors, and psychosocial factors), and health system factors [21]. We included the following as main control variables: sex (male, female), age (18–24, 25–29, 30–34, 35–39, 40–44, 45–59 years) (representing biological factors), nationality (Syrian, Iraqi, Afghan) (representing factors referring to the country of origin), partnership status (never married, married and living with partner, married and not living with partner, married and no information on place of residence of partner, widowed or divorced, or no information on partner status) (representing psychosocial factors), and education (low level (International Standard Classification of Education (ISCED) 0–1) or no information on education, medium level (ISCED 2), high level (ISCED 3–6)) (representing socioeconomic factors). These socio-demographic variables were cited in previous studies as crucial determinants of SRH [42, 43]. For some of our multivariate analyses, we also consider migration-specific characteristics, such as the length of stay (0–18, 19–24, 25–30, 31–36, 37 or more months) and the length of the asylum application process (0–3, 4–6, 7–14, 15 or more months; decision still open; no information on length of asylum process), which represent health system and psychosocial factors.

## Statistical analysis methods

The analyses consist of three steps. First, we provide descriptive results on the share of interviewed refugees in (very) good self-rated health (vgSRH) in the two countries. Second, we explore determinants for SRH. Separately for the two countries, probit regression models with SRH as a dependent variable are used to estimate average marginal effects (AME). These AME represent the average effects of a variable on the probability of perceiving one's health as (very) good, and are comparable across different models [44]. Positive/negative coefficients indicate that the corresponding group had vgSRH more/less often than the reference group.

Third, we investigate whether the initially limited access of AS&R to health services in Germany is associated with differences in SRH in Germany and Austria. Matching estimators are used compare the outcomes (i.e., SRH) of individuals who are as similar as possible with the sole exception of their treatment status. Whereas in medical studies "treatment" typically refers to the introduction of a new drug or a new surgical procedure [45, 46], we define treatment as AS&R being given unlimited access to health services in the host country from the time of arrival onward. An individual's treatment status is equal to one if s/he is residing in Austria, and to zero if s/he is residing in Germany. The efficacy of the treatment is estimated via the average treatment effect (ATE) of those receiving it. Within the counterfactual framework [47, 48], we denote $Y_0$ as the observed outcome if a subject did not receive treatment, and $Y_1$ as the counterfactual for that subject if s/he was exposed to treatment. For a subject who received treatment, we denote $Y_1$ as the observed outcome, and $Y_0$ as the counterfactual outcome. To address this missing data problem, the Stata software package provides methods for estimating treatment parameters like the ATE, which is the mean of the difference between the observed and the counterfactual outcome: $ATE = E(Y_1 - Y_0)$. We perform a five-nearest-neighbor matching procedure, and apply matching with replacement, which increases matching quality and decreases bias [49]. The caliper width for valid matches [50] is set to 0.3.

Propensity score matching is used to control for differences between the two countries in the structure of the AS&R population [51]. Propensity scores are the conditional probability of assignment to treatment (i.e., residing in Austria) given a vector of observed covariates (sex, nationality, age, partnership status, and education) [52]. As implemented in the Stata software via the command *teffects*, after conditioning on these covariates, any remaining influences on the treatment are not related to the potential outcome [53]. Statistical analyses were conducted in Stata version 15 [53].

## Ethics

The ReHIS was approved by the research commission of the Vienna University of Economics and Business. The "Ethical Guidelines for Good Research Practice" issued by the Oxford Refugee Studies Centre [54] were fully adhered to. Participants provided their informed consent to participate in the study. Because the survey was conducted via CATIs, interviewee consent was not documented, as only those participants who gave their explicit consent were interviewed.

The authors used only de-identified data from the ReHIS and the IAB-BAMF-SOEP-Refugee Survey 2016, and were thus exempt from IRB review. Consent was obtained by providing all participants with a declaration of data protection indicating that participation was voluntary, and identities would be kept confidential.

## Results

### General characteristics

The respondents in our German analytical sample were predominantly male, and had a mean age of 33 years. The majority were Syrian (Table 1). A large share were married and living with

**Table 1. Sample characteristics of AS&R and share in vgSRH, by country.**

| | Sample characteristics | | Share in vgSRH | | t-Test |
| --- | --- | --- | --- | --- | --- |
| | | | mean (95%CI) | | |
| | Germany | Austria | Germany | Austria | |
| Sex | | | | | |
| Male | 64% | 87% | 0.76 (0.74; 0.78) | 0.90 (0.87; 0.93) | *** |
| Female | 36% | 13% | 0.65 (0.62; 0.68) | 0.79 (0.67; 0.91) | + |
| Nationality | | | | | |
| Syria | 67% | 62% | 0.74 (0.72; 0.76) | 0.93 (0.90; 0.96) | *** |
| Iraq | 17% | 17% | 0.68 (0.63; 0.72) | 0.88 (0.79; 0.96) | *** |
| Afghanistan | 16% | 21% | 0.66 (0.61; 0.70) | 0.75 (0.65; 0.85) | + |
| Age | | | | | |
| 18–24 years | 22% | 26% | 0.84 (0.81; 0.87) | 0.87 (0.80; 0.94) | |
| 25–29 years | 16% | 24% | 0.75 (0.71; 0.79) | 0.95 (0.90; 0.99) | *** |
| 30–34 years | 18% | 18% | 0.76 (0.72; 0.80) | 0.89 (0.82; 0.97) | * |
| 35–39 years | 17% | 15% | 0.70 (0.66; 0.74) | 0.89 (0.81; 0.98) | ** |
| 40–44 years | 11% | 9% | 0.69 (0.64; 0.74) | 0.94 (0.86; 1.02) | ** |
| 45–59 years | 15% | 8% | 0.49 (0.45; 0.54) | 0.67 (0.49; 0.85) | + |
| Mean age in years | 33 | 31 | | | |
| Partnership status | | | | | |
| Never married | 27% | 53% | 0.83 (0.80; 0.86) | 0.89 (0.85; 0.94) | * |
| Married, living with partner | 59% | 26% | 0.70 (0.68; 0.72) | 0.91 (0.85; 0.97) | *** |
| Married, not living with partner | 10% | 8% | 0.65 (0.59; 0.71) | 0.77 (0.62; 0.93) | |
| Married, no information on partner | 1% | 10% | 0.69 (0.43; 0.94) | 0.89 (0.79; 1.00) | + |
| Widowed/divorced/no answer | 4% | 3% | 0.44 (0.35; 0.53) | 0.85 (0.62; 1.07) | ** |
| Education | | | | | |
| Low level (ISCED 0–1) or no answer | 43% | 25% | 0.66 (0.63; 0.68) | 0.82 (0.74; 0.90) | ** |
| Medium level (ISCED 2) | 19% | 13% | 0.75 (0.72; 0.79) | 0.86 (0.76; 0.96) | + |
| High level (ISCED 3–6) | 38% | 62% | 0.77 (0.74; 0.79) | 0.92 (0.88; 0.95) | *** |
| Length of stay | | | | | |
| 0–18 months | 70% | 3% | 0.72 (0.70; 0.74) | 0.64 (0.30; 0.98) | |
| 19–24 months | 11% | 6% | 0.72 (0.67; 0.77) | 0.82 (0.64; 0.99) | |
| 25–30 months | 9% | 17% | 0.73 (0.67; 0.78) | 0.83 (0.73; 0.92) | |
| 31–36 months | 5% | 39% | 0.71 (0.64; 0.79) | 0.92 (0.87; 0.96) | *** |
| 37 months and more | 5% | 36% | 0.67 (0.59; 0.75) | 0.91 (0.86; 0.96) | *** |
| Length of asylum process | | | | | |
| 0–3 months | 23% | 16% | 0.73 (0.70; 0.77) | 0.88 (0.80; 0.97) | ** |
| 4–6 months | 18% | 13% | 0.73 (0.69; 0.77) | 0.89 (0.80; 0.99) | * |
| 7–14 months | 20% | 40% | 0.76 (0.72; 0.79) | 0.92 (0.88; 0.96) | *** |
| 15 months and more | 6% | 27% | 0.77 (0.70; 0.83) | 0.83 (0.76; 0.91) | |
| Decision still open | 29% | 4% | 0.66 (0.63; 0.69) | 0.86 (0.65; 1.07) | |
| No information | 4% | 0% | 0.73 (0.65; 0.81) | | |
| Total | 100% | 100% | 0.72 (0.70; 0.73) | 0.89 (0.85; 0.92) | *** |
| Total (N) | 2,854 | 374 | 2,854 | 374 | |

Sources: IAB-BAMF-SOEP 2016, ReHIS

Significance levels:

+ p<0.10

* p<0.05

** p<0.01

*** p<0.001.

Note: AS&R: asylum seekers and refugees, vgSRH: (very) good self-rated health.

a partner, while smaller shares had never been married or were married and not living with their partner. Very few of the respondents were divorced or widowed, or provided no information on their partnership status. Roughly four out of 10 respondents reported having either a low level of education (ISCED 0–1; 37%) or provided no answer (6%), while roughly the same share reported having a high level of education (ISCED 3–6), and two out of 10 said they had a medium level of education (ISCED 2). A large share of the interviewees had arrived in Germany within 18 months of the survey (conducted in 2016). One out of four respondents reported that the duration of their asylum application process was three months or less, and was thus rather short. Two out of four indicated that they had waited 4–6 months for the decision, and two out of 10 said they had waited 7–14 months. Three out of 10 of the respondents reported that the decision regarding their asylum application was still open at the time of the interview.

The gender distribution in our Austrian analytical sample was also unbalanced (87% males). The majority of the respondents were Syrian, and the mean age in the sample was 31 years. More than one half of the respondents had never been married; one out of four were married and living with their partner; 8% were married and were not living with their partner at the time of the interview; and a substantial share were married, but reported no information on their partner's place of residence. A majority of the respondents had a high level of education (62%), while smaller shares had a low level of education or provided no information on their educational level (13% and 12% respectively, totaling to 25%), or had a medium level of education. More than one-third of the respondents had been living in Austria for more than three years, and four out of 10 had been living there for 31–36 months. Therefore, three out of four respondents in the Austrian cohort had been living in the host country for more than two and a half years when they were interviewed in 2018. The refugee status of almost all of the Austrian respondents had been officially recognized, with only 4% reporting that they were still waiting for a decision on their asylum application. Three out of 10 of the respondents reported that the length of their asylum application process had been relatively short (six months or less), while a larger share said they had received a decision on their application within 7–14 months.

## Comparison of SRH in Germany and Austria

The shares of AS&R respondents who had vgSRH was smaller in Germany (72%) than in Austria (89%) (Table 1). The difference in the vgSRH levels in the two countries was highly statistically significant (p<0.001).

In both countries, males were more likely than females to have vgSRH (76% versus 65% in Germany; 90% versus 79% in Austria). The differences by nationality were substantial: the shares of respondents who had vgSRH were higher among Syrians (74% in Germany; 93% in Austria) than among Iraqis (68% in Germany; 88% in Austria) or Afghans (66% in Germany; 75% in Austria). There were also large differences in self-reported health by age in Germany: 84% of young people aged 18–24, but only 49% of those aged 45–59, had vgSRH. The variation by age was less pronounced in Austria, where the respondents aged 35–39 (89%) and aged 40–44 (94%) were most likely to have vgSRH. In Austria, married people who were not living with their partner had comparably poor health (77% with vgSRH), whereas most of the respondents with other partnership statuses had vgSRH (85%-91%). In Germany, by contrast, the level of SRH was highest among those who had never been married (83% with vgSRH), and was lowest among those who were widowed or divorced (44% with vgSRH). As expected, education was associated with SRH, with the respondents with higher educational levels being most likely to

have vgSRH. Due to overlapping confidence intervals (Table 1), the health differences within the two countries lack statistical significance for some of the analyzed covariates.

T-tests indicated that the differences between the AS&R in Germany and Austria were statistically significant for males (76% versus 90%). This was also found to be the case for Syrian nationals (74% versus 93%) and for Iraqi nationals (68% versus 88%); for the majority of age groups; and for various partnership groups, such as married people who were living with their partner (70% versus 91%). Furthermore, the differences between the AS&R in the two countries were statistically significant for those with low educational levels (66% versus 82%) and with high educational levels (77% versus 92%), and for various groups based on the length of their stay and the length of their asylum application process. All of the statistically significant differences indicate that the share of AS&R who had vgSRH was lower in Germany than in Austria.

## Probit regressions

Among the AS&R in Germany, sex, nationality, age, and education were found to be significantly associated with vgSRH: men, Syrians, younger individuals, and those with higher levels of education were significantly more likely than other groups to have vgSRH (Table 2, Model 1 for Germany). Looking at partnership status, we can see that the respondents who were widowed or divorced, or provided no information on their partner status, were less likely than those who were married or had never been married to have vgSRH, but the differences were not statistically significant. In Austria, Afghans and people aged 45–59 were significantly less likely than other groups to have vgSRH. The estimated coefficients were also statistically significant at the 10% level for females, for the 30–34 age group, and for those with high levels of education (Table 2, Model 1 for Austria). Overall, the effects were found to be similar in Germany and Austria, with the exception that being divorced or widowed was shown to be detrimental in Germany, but not in Austria.

Adjustment for the length of the asylum application process (Table 2, Model 2) or the length of stay (Table 2, Model 3) did not mediate the differences in the likelihood of having vgSRH, with the exception that in Germany, individuals with an asylum application process that lasted 15 months or longer were significantly more likely to have vgSRH. Fig 1 illustrates the average marginal effects of the socio-demographic control variables included in the analysis.

## Propensity score matching

To assess the differences in SRH between the AS&R in Germany and Austria, we performed propensity score matching (PSM), and estimated the ATE (see Table 3 for PSM specifications). As we mentioned earlier, PSM was used to identify the AS&R with similar characteristics (in terms of age, sex, nationality, education, and partnership status) in Austria and Germany, and thus allowed us to estimate non-confounded remaining country effects [52].

The matched sample for the comparative analysis consisted of 374 refugees in Austria and 506 refugees in Germany (Table 3). The estimated ATE was 0.12 (Table 3). This indicates that the probability of the AS&R having vgSRH was, on average, 12% higher in Austria than in Germany.

The characteristics of the matched sample (Table 4) indicated that unlike in the unmatched sample, there was a rough convergence of matching variables (Table 1). To assess the matching quality and the bias in the estimation of the causal effect, we provided the mean bias, LR chi$^2$, before and after matching and Rosenbaum bounds were applied (Table 3). The model specification showed a mean bias of 3.3% (i.e., the relative difference between the matched samples

**Table 2. Average marginal effects (and 95%CI) for vgSRH, by country.**

| | Model 1 | Model 1 | Model 2 | Model 2 | Model 3 | Model 3 |
|---|---|---|---|---|---|---|
| | Germany | Austria | Germany | Austria | Germany | Austria |
| Sex | | | | | | |
| Male (ref.) | 0 | 0 | 0 | 0 | 0 | 0 |
| Female | -0.08*** | -0.11+ | -0.09*** | -0.10+ | -0.08*** | -0.07 |
| | (-0.12; -0.05) | (-0.23; 0.01) | (-0.12; -0.05) | (-0.22; 0.01) | (-0.12; -0.05) | (-0.18; 0.04) |
| Nationality | | | | | | |
| Syria (ref.) | 0 | 0 | 0 | 0 | 0 | 0 |
| Iraq | -0.06** | -0.05 | -0.04+ | -0.05 | -0.06** | -0.06 |
| | (-0.10; -0.01) | (-0.14; 0.03) | (-0.09; 0.00) | (-0.14; 0.03) | (-0.10; -0.01) | (-0.15; 0.02) |
| Afghanistan | -0.09*** | -0.18** | -0.07** | -0.19** | -0.09*** | -0.19*** |
| | (-0.13; -0.04) | (-0.29; -0.07) | (-0.12; -0.02) | (-0.31; -0.06) | (-0.14; -0.04) | (-0.30; -0.08) |
| Age | | | | | | |
| 18–24 years (ref.) | 0 | 0 | 0 | 0 | 0 | 0 |
| 25–29 years | -0.07** | -0.00 | -0.07** | -0.00 | -0.07** | -0.02 |
| | (-0.12; -0.02) | (-0.07; 0.06) | (-0.12; -0.03) | (-0.07; 0.07) | (-0.12; -0.02) | (-0.08; 0.05) |
| 30–34 years | -0.06* | -0.10+ | -0.07* | -0.09+ | -0.06* | -0.11* |
| | (-0.12; -0.01) | (-0.20; 0.01) | (-0.12; -0.02) | (-0.20; 0.01) | (-0.12; -0.01) | (-0.21; -0.00) |
| 35–39 years | -0.13*** | -0.07 | -0.13*** | -0.07 | -0.13*** | -0.09+ |
| | (-0.18; -0.07) | (-0.18; 0.03) | (-0.19; -0.07) | (-0.17; 0.04) | (-0.18; -0.07) | (-0.20; 0.02) |
| 40–44 years | -0.15*** | -0.09 | -0.15*** | -0.08 | -0.15*** | -0.10 |
| | (-0.21; -0.08) | (-0.26; 0.08) | (-0.22; -0.09) | (-0.24; 0.09) | (-0.21; -0.08) | (-0.27; 0.07) |
| 45–59 years | -0.33*** | -0.40*** | -0.34*** | -0.40*** | -0.33*** | -0.40*** |
| | (-0.40; -0.27) | (-0.59; -0.20) | (-0.40; -0.28) | (-0.59; -0.20) | (-0.40; -0.27) | (-0.59; -0.21) |
| Partnership status | | | | | | |
| Never married (ref.) | 0 | 0 | 0 | 0 | 0 | 0 |
| Married, living with partner | 0.00 | 0.06 | 0.01 | 0.06 | 0.00 | 0.06 |
| | (-0.05; 0.06) | (-0.03; 0.14) | (-0.04; 0.06) | (-0.03; 0.14) | (-0.05; 0.06) | (-0.02; 0.15) |
| Married, not living with partner | -0.05 | -0.03 | -0.05 | -0.03 | -0.06 | -0.02 |
| | (-0.12; 0.01) | (-0.17; 0.11) | (-0.12; 0.02) | (-0.17; 0.11) | (-0.13; 0.01) | (-0.15; 0.11) |
| Married, no information on partner | 0.09 | 0.05 | 0.09 | 0.05 | 0.08 | 0.05 |
| | (-0.09; 0.26) | (-0.05; 0.16) | (-0.09; 0.26) | (-0.06; 0.15) | (-0.09; 0.26) | (-0.05; 0.16) |
| Widowed/divorced/no answer | -0.19*** | 0.08 | -0.19*** | 0.07 | -0.19*** | 0.07 |
| | (-0.29; -0.09) | (-0.04; 0.20) | (-0.29; -0.09) | (-0.05; 0.19) | (-0.29; -0.09) | (-0.05; 0.19) |
| Education | | | | | | |
| Low level (ISCED 0–1) or n.a. (ref.) | 0 | 0 | 0 | 0 | 0 | 0 |
| Medium level (ISCED 2) | 0.06* | 0.01 | 0.05* | 0.01 | 0.05* | -0.01 |
| | (0.01; 0.10) | (-0.10; 0.13) | (0.01; 0.10) | (-0.11; 0.13) | (0.01; 0.10) | (-0.13; 0.11) |
| High level (ISCED 3–6) | 0.09*** | 0.08+ | 0.09*** | 0.08+ | 0.09*** | 0.07+ |
| | (0.05; 0.13) | (-0.01; 0.17) | (0.05; 0.12) | (-0.01; 0.17) | (0.05; 0.13) | (-0.01; 0.16) |
| Length of asylum process | | | | | | |
| 0–3 months (ref.) | | | 0 | 0 | | |
| 4–6 months | | | -0.01 | -0.01 | | |
| | | | (-0.06; 0.04) | (-0.14; 0.13) | | |
| 7–14 months | | | 0.04 | 0.04 | | |
| | | | (-0.01; 0.09) | (-0.05; 0.14) | | |
| 15 months and more | | | 0.08* | 0.03 | | |
| | | | (0.01; 0.15) | (-0.08; 0.14) | | |

*(Continued)*

**Table 2.** (Continued)

| | Model 1 | Model 1 | Model 2 | Model 2 | Model 3 | Model 3 |
|---|---|---|---|---|---|---|
| | **Germany** | **Austria** | **Germany** | **Austria** | **Germany** | **Austria** |
| Decision still open | | | -0.04 | 0.04 | | |
| | | | (-0.09; 0.01) | (-0.12; 0.20) | | |
| No information | | | 0.06 | | | |
| | | | (-0.02; 0.13) | | | |
| Length of stay | | | | | | |
| 0–18 months | | | | | 0 | 0 |
| 19–24 months | | | | | -0.02 | 0.09 |
| | | | | | (-0.07; 0.03) | (-0.21; 0.39) |
| 25–30 months | | | | | 0.01 | 0.14 |
| | | | | | (-0.04; 0.07) | (-0.12; 0.41) |
| 31–36 months | | | | | 0.04 | 0.22 |
| | | | | | (-0.03; 0.11) | (-0.04; 0.47) |
| 37 months and more | | | | | -0.00 | 0.21 |
| | | | | | (-0.07; 0.07) | (-0.05; 0.47) |
| N | 2,854 | 374 | 2,854 | 374 | 2,854 | 374 |

Sources: IAB-BAMF-SOEP 2016, ReHIS

Significance levels:

+ p<0.10

* p<0.05

** p<0.01

*** p<0.001.

Note: vgSRH: (very) good self-rated health.

across all included covariates), and thus indicated a good match [55]. When we compared LR chi$^2$ before and after matching, we found that after matching, the covariates no longer predicted group assignment [56]. The Rosenbaum bounds strategy was used to assess the potential impact of hidden bias; i.e., the bias arising from confounding variables that were simultaneously associated with the treatment variable and the outcome variable [57]. This approach enabled us to obtain a high level of matching quality. The use of other specifications (regarding caliper width, number of nearest neighbors, matching variables) resulted in very similar ATEs (results available upon request), but the goodness of fit parameters were lower. We discarded the length of stay and the length of the asylum application process as matching characteristics due to the sample composition; however, these characteristics certainly can be included in future studies.

## Discussion

The vast majority of the AS&R who were interviewed after arriving in Germany and Austria in recent years rated their health as (very) good. The vgSRH proportions of 72% found in Germany and of 89% found in Austria exceeded those reported in earlier studies in the high-income countries of Germany and the Netherlands [12, 22], which might be attributable to period effects or to differences in sample compositions.

While the overall health ratings in our sample were positive, SRH varied considerably by age and education, confirming that certain health determinants of non-migrant populations

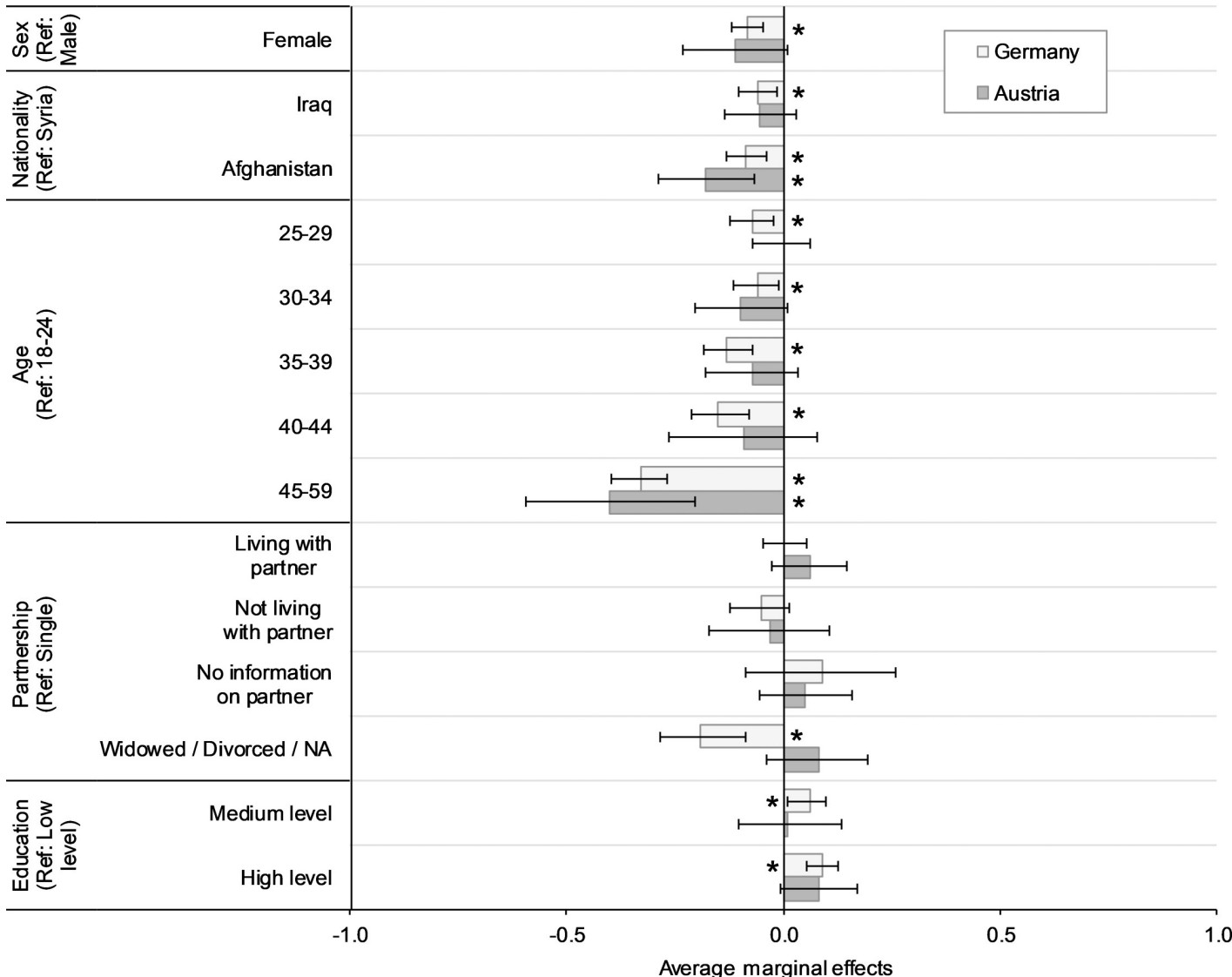

**Fig 1. Average marginal effects for sex, nationality, age, partnership status, and education; by country.** Sources: IAB-BAMF-SOEP 2016, ReHIS; Remark: Average marginal effects as estimated in model 1 in Table 2. Asterisks denote significant within-country differences compared to the reference group (Ref) (p<0.05).

**Table 3. Model specifications and outcome of propensity score matching.**

| Criterion | Value |
|---|---|
| Matching variables | Sex, nation, age group, partnership status, education |
| Maximum number of nearest neighbors | 5 |
| Caliper width | 0.3 |
| Number of matched individuals in Germany | 506 |
| Number of matched individuals in Austria | 374 |
| Mean bias | 3.3 |
| LR chi$^2$ | 346.95 (p<0.001) before matching; 5.40 (p = 0.979) after matching |
| Rosenbaum's bounds Γ | 2.7 (p = 0.031)– 2.8 (p = 0.052) |
| **ATE (95%CI)** | **0.12 (0.04; 0.20)** |

**Table 4. Characteristics of matched sample, by country.**

| | Sample characteristics | | Share in vgSRH mean (95%CI) | | t-Test |
|---|---|---|---|---|---|
| | Germany | Austria | Germany | Austria | |
| Sex | | | | | |
| Male | 77% | 87% | 0.76 (0.71; 0.80) | 0.90 (0.87; 0.93) | *** |
| Female | 23% | 13% | 0.61 (0.52; 0.70) | 0.79 (0.67; 0.91) | + |
| Nationality | | | | | |
| Syria | 56% | 62% | 0.75 (0.70; 0.80) | 0.93 (0.90; 0.96) | *** |
| Iraq | 23% | 17% | 0.68 (0.58; 0.76) | 0.88 (0.79; 0.96) | ** |
| Afghanistan | 21% | 21% | 0.70 (0.62; 0.79) | 0.75 (0.65; 0.85) | + |
| Age | | | | | |
| 18–24 years | 19% | 26% | 0.84 (0.77; 0.92) | 0.87 (0.80; 0.94) | |
| 25–29 years | 22% | 24% | 0.73 (0.65; 0.82) | 0.95 (0.82; 0.97) | *** |
| 30–34 years | 19% | 18% | 0.80 (0.71; 0.88) | 0.89 (0.82; 0.97) | + |
| 35–39 years | 18% | 15% | 0.65 (0.55; 0.75) | 0.89 (0.81; 0.98) | ** |
| 40–44 years | 10% | 9% | 0.75 (0.62; 0.87) | 0.94 (0.86; 1.02) | ** |
| 45–59 years | 11% | 8% | 0.48 (0.35; 0.62) | 0.67 (0.49; 0.85) | |
| Mean age in years | 33 | 31 | | | |
| Partnership status | | | | | |
| Never married | 34% | 53% | 0.82 (0.76; 0.87) | 0.89 (0.85; 0.94) | * |
| Married, living with partner | 26% | 26% | 0.70 (0.63; 0.76) | 0.91 (0.85; 0.97) | *** |
| Married, not living with partner | 8% | 8% | 0.75 (0.67; 0.84) | 0.77 (0.62; 0.93) | |
| Married, no information on partner | 10% | 10% | 0.67 (0.35; 0.98) | 0.89 (0.79; 1.00) | + |
| Widowed/divorced/no answer | 3% | 3% | 0.36 (0.20; 0.53) | 0.85 (0.62; 1.07) | ** |
| Education | | | | | |
| Low level (ISCED 0–1) or no answer | 32% | 25% | 0.68 (0.61; 0.75) | 0.82 (0.74; 0.90) | ** |
| Medium level (ISCED 2) | 20% | 13% | 0.76 (0.68; 0.85) | 0.86 (0.76; 0.96) | |
| High level (ISCED 3–6) | 48% | 62% | 0.73 (0.68; 0.79) | 0.92 (0.88; 0.95) | *** |
| Length of stay | | | | | |
| 0–18 months | 72% | 3% | 0.72 (0.67; 0.76) | 0.64 (0.30; 0.98) | |
| 19–24 months | 10% | 6% | 0.73 (0.62; 0.84) | 0.82 (0.64; 0.99) | |
| 25–30 months | 8% | 17% | 0.77 (0.65; 0.89) | 0.83 (0.73; 0.92) | |
| 31–36 months | 5% | 39% | 0.81 (0.60; 1.03) | 0.92 (0.87; 0.96) | |
| 37 months and more | 5% | 36% | 0.65 (0.44; 0.86) | 0.91 (0.86; 0.96) | *** |
| Length of asylum process | | | | | |
| 0–3 months | 21% | 16% | 0.78 (0.70; 0.86) | 0.88 (0.80; 0.97) | |
| 4–6 months | 16% | 13% | 0.74 (0.64; 0.83) | 0.89 (0.80; 0.99) | * |
| 7–14 months | 18% | 40% | 0.76 (0.67; 0.85) | 0.92 (0.88; 0.96) | *** |
| 15 months and more | 7% | 27% | 0.76 (0.61; 0.90) | 0.83 (0.76; 0.91) | |
| Decision still open | 34% | 4% | 0.64 (0.57; 0.72) | 0.86 (0.65; 1.07) | |
| No information | 5% | 0% | 0.77 (0.58; 0.96) | | |
| Total | 100% | 100% | 0.72 (0.68; 0.76) | 0.89 (0.85; 0.92) | *** |
| Total (N) | 506 | 374 | 506 | 374 | |

Sources: IAB-BAMF-SOEP 2016, ReHIS

Significance levels:

+ p<0.10

* p<0.05

** p<0.01

*** p<0.001.

Note: vgSRH: (very) good self-rated health.

also apply to the AS&R population. Significant sex differences in favor of men were found only in Germany, but displayed the same tendency in Austria.

Refugee-specific characteristics were found to have almost no significant impact, which is not in line with the findings of previous studies [26, 27]. The length of stay (LOS) in the host country and the length of the asylum application process turned out to be of minor importance for SRH. These findings may be explained by large within-group heterogeneity in terms of health. For example, in Germany, the asylum application process is shorter for individuals with special needs, such as a disability. Thus, having a longer LOS might be associated with being better integrated into the host society, but it can also lead to an accumulation of economic and social disadvantages [58]. Both outcomes are associated with health [20, 21, 59].

The health differences found by country of origin–in Germany and Austria, Syrians had the highest levels of SRH; while in Austria, Afghans had particularly low levels of SRH–point to different trajectories over time. Additionally, these patterns may reflect country-specific values toward and processes of marginalization of subgroups of AS&R [60–63], and they may indicate the interdependency of origin-related and host-country-specific conditions.

The AS&R surveyed in Germany assessed their health as being worse than those surveyed in Austria. This difference could be only partially explained by compositional differences. Balancing the samples in terms of age, sex, education, nationality, and partnership status, the probability of having vgSRH was found to be 12% lower for the AS&R in Germany than for the AS&R in Austria. However, this finding might be driven by several limitations, as discussed below.

## Limitations and strengths

First, although the sample was balanced with PSM, unobserved heterogeneity across the samples cannot be ruled out; e.g., in terms of social and economic integration, health needs, and initial and migration-related circumstances. Nevertheless, we found no evidence of large differences in these characteristics in the German and the Austrian samples. Moreover, the decision of the AS&R to settle in Austria (and not to move on to Germany) might have been driven by negative health selection, i.e., those with poorer health remained in Austria due to their state of health [64]; or by positive causation, i.e., those who remained in Austria might have experienced a slightly shorter and less exhausting journey, which would be associated with better health outcomes [22].

Second, host country-specific characteristics at the societal levels–such as offers of support, integration measures, perceptions of minorities, experiences of discrimination or segregation, and ethnic networks–might contribute to the differences in the health assessments of the AS&R in Austria and Germany [62]. However, it is important to keep in mind that Germany and Austria are culturally very closely aligned, with similar languages and similar attitudes toward AS&R [65, 66].

Third, the initially limited access to health care that the AS&R in Germany experienced might partly explain the differences. Up to 15 months after they arrive, AS&R in Germany receive only basic medical treatment, which might exacerbate their unmet health needs. Earlier findings reported that AS&R in Germany often have problems accessing psychiatric care and medical treatment [67].

Fourth, as the surveys were conducted at an interval of two years, there may have been period effects; i.e., whether and, if so, to what extent the health assessments of the AS&R in the years 2016 and 2018 differed should be considered. Both the actual changes in health between 2016 and 2018 and indirect effects–e.g., changes in the attitudes of the majority population toward AS&R [68] or the integration processes of AS&R [69, 70]–might have influenced the

respondents' health assessments. These effects might partially account for the health differences found among the AS&R in Germany and Austria.

Fifth, the instruments and data collection of the IAB-BAMF-SOEP 2016 and the ReHIS differed, which might have influenced response patterns. Compared to the ReHIS questionnaire, the IAB-BAMF-SOEP questionnaire was much more comprehensive. For example, in the IAB-BAMF-SOEP, the core household questionnaire (100 questions) was designed to last 15 minutes, and each personal questionnaire (450 questions) was designed to take an additional 30 minutes [35, 71]. The face-to-face interviews conducted in the IAB-BAMF-SOEP lasted 28 minutes (first percentile) to 250 minutes (99th percentile), with a median of 83 minutes, and differed by self-rated health (vgSRH: median of 81 minutes, not good SRH: median of 87 minutes). The question regarding SRH followed questions regarding personal characteristics and migration history, which may have caused a halo effect. The interviews conducted in the ReHIS lasted nine minutes (first percentile) to 60 minutes (99th percentile), with a median of 19 minutes. The question regarding SRH was asked almost at the beginning. The length of the interview differed by self-rated health (vgSRH: median of 18 minutes, not good SRH: median of 21 minutes). Thus, the different approaches to data collection used in the surveys (length of interviews, structure of the questionnaires, and interview mode (IAB-BAMF-SOEP: CAPI face-to-face-interviews, ReHIS: telephone interviews)) might have resulted in different forms of measurement bias, and might have biased the answers.

Sixth, there are limitations in the representativeness of the samples. Overall, the selectivity of respondents is a well-known issue that arises when conducting refugee surveys [72–74]. In their respective country contexts, the ReHIS and the IAB-BAMF-SOEP-Refugee Survey 2016 were among the first surveys to focus on the recently arrived AS&R population from Syria, Iraq, and Afghanistan. However, neither survey sample was fully representative of the national AS&R population [2, 40]. It might be assumed that AS&R with lower levels of education and poor health were underrepresented in our analysis [75, 76]. Moreover, the sample was unbalanced in terms of sex and host country; the majority were male and lived in Germany. These imbalances do neither reflect the population of refugees or locals in the respective countries nor a proportionality of AS&R in Germany and Austria. In 2016, 34% of AS&R in Germany [77] and 33% of AS&R in Austria [78] were female; however, female refugees are still less researched and underrepresented in surveys [79]. The number of AS&R in Germany was more than nine times higher than in Austria [1]. The (disproportionally) higher number of AS&R in our German sample was based on the larger sample in the IAB-BAMF-SOEP-Refugee Survey 2016. The analyses were adjusted for gender and host country, i.e. these imbalances do not bias the results. The propensity score matching allows imbalances to be compensated.

To evaluate how, for example, the different educational profiles influenced our results, we calculated weights adjusting for education. After applying these weights, the share of individuals who had vgSRH decreased only slightly, from 89% to 86% in Austria (results available on request). The high proportion of the AS&R in the German sample who had a lower level of education cannot be fully explained, and country-specific or education-specific self-selection into the surveys cannot be ruled out. After analyzing educational differences by country and sex (see S1 Fig), we found that the Syrians and Iraqis in Austria reported having substantially higher levels of education than their counterparts in Germany. We also found significant differences between men and women among the Iraqi and Afghan AS&R in Germany, with lower shares of women than men reporting a high level of education. However, after our models were adjusted for education, sex, and nationality, these compositional differences did not explain the differences in the SRH of AS&R in Austria and Germany. To minimize language barriers and to ensure that all of the participants understood the questions–and, thus, to minimize the educational bias–both of the questionnaires were subject to pretests before the data

collection. Moreover, qualified interviewers conducted the interviews, and questionnaires in several languages (translated and harmonized by two translators) were made available during the interviews. In addition, audiovisual tools and aids were applied in the IAB-BAMF-SOEP survey.

Seventh, the data provide comparable information on health and individual characteristics, but do not cover the full set of possible health determinants among the AS&R. To improve comparability, only a small set of socio-demographic characteristics, as derived from the Social Determinants of Health framework, were integrated into the analyses. To achieve comparability between two samples, we applied PSM, which is appropriate for estimating non-confounded effects [52]. Further relevant determinants, as elaborated in earlier studies [80], might be addressed in future studies.

Eighth, the focus of this paper was on SRH, which represents perceived, but not a medically certified health. SRH is based on self-disclosure, and does not cover all elements of health. It does not provide information on special health circumstances, and it might be driven by subjective short-term influences, as well as by external and internal differences in assessments of and responses to this question. Nonetheless, SHR has been verified as a useful and valid summary of perceived overall health [81–83] that includes both somatic and physical health [84, 85]. Our data indicate that there are strong correlations between SRH and mental health (e.g., depressiveness, Chi$^2$: p<0.001), as well as between SRH and physical health (e.g., frequency of physical pain, Chi$^2$: p<0.001). Thus, in our sample, it is not possible to differentiate the impact of psychiatric and somatic health on the assessment of SRH. As the AS&R had traumatizing reasons for fleeing, and were having to manage post-displacement stressors [86, 87], they frequently reported mental health problems [7, 88]. Thus, SRH might reflect psychiatric health. Both data sources include question related to mental health, but available data do not allow to generate in both surveys standard scales for mental health, like EURO-D scale or Kessler-10 scale [89, 90]. Another caveat is the fact that experience of violence and torture, which are crucial for refugee health, are nor captured in the two surveys. However, compared to a mere mental health assessment, self-rated health is less subject to bias [91]. We assume a simultaneity and an interaction of the two health areas, and interpret the results in terms of the general health status of individuals. Moreover, previous research has suggested that SRH is sensitive to cultural differences [92, 93], and that SHR responses depend in part on the interview language and on the translation of that language [94]. However, these effects are unlikely to explain the health differences found among the AS&R in Germany and Austria. Subsequent studies could address other health dimensions or specific health determinants.

Finally, cultural differences in self-reported health are relevant [95, 96], but are not further explored in this paper.

The great strength of our study is that it provides a comparative perspective on health differences and health mechanisms in two neighboring, culturally similar, high-income European countries that have experienced high levels of AS&R immigration in recent years. Our approach allowed us to elaborate general determinants and country-specific differences for three nationalities of one refugee cohort (who immigrated between 2013 and 2016). However, future studies might consider additional countries in order to analyze the impact of nationally diverse health policies and settings, or to take the internal heterogeneity of the AS&R population into account. While previous studies on this population for Europe mainly focused on the mental health of refugees [97, 98], this study provides findings on the general subjective health of AS&R. While mental health is a major concern for AS&R from conflict regions [98, 99], the general health of this population should not be neglected. Assessments that cover mental health only are not comprehensive, and could lead to an overestimation of the health challenges associated with refugee immigration.

## Conclusions

When assessing the health levels in a society, AS&R represent a particularly vulnerable group. Our results do not indicate that the general health needs of AS&R are greater than those of the non-migrant population. Nevertheless, the SRH levels within the AS&R population vary considerably. As women, older refugees, and refugees with lower levels of education report having worse health than other groups, the needs of these groups in particular should be addressed by health-promoting measures. As these determinants correspond to those in the non-migrant population, similar strategies are conceivable. For example, comprehensive care, including more frequent screenings and better professional health advice, could be offered for some groups. Additionally, as the Afghan refugees in our sample reported having lower levels of health than other nationality groups, there may be a need for nationality-specific and culturally sensitive treatments and health services. While we cannot clearly identify the causes of this poorer health assessment, previous studies have highlighted the multidimensionality of health risks [100], which act at different levels. In terms of individual characteristics, language abilities, institutionalized knowledge, and exchange networks are important health resources [7, 25]. Thus, promoting these resources might lead to improvements in health.

Moreover, the results of our analyses are in line with the known differences between the two countries in access health care, as the AS&R in our sample reported having better health in Austria than in Germany. Although Germany and Austria have very similar healthcare delivery systems in terms of health expenditures and the density of practitioners [28], the German model has more barriers to initial access for AS&R. This lack of access may be associated with long-lasting unmet health needs, poorer health, and higher public health expenditures [101–104]. Thus, the health of AS&R and health systems in general may be improved by removing barriers to accessing health services.

## Supporting information

**S1 Fig. Educational level by nationality and sex, by country.** Sources: IAB-BAMF-SOEP 2016, ReHIS.
(TIF)

## Acknowledgments

We thank our study participants and field staff; without their support, the implementation of this study would have been impossible. We are grateful for the cooperation with the FIMAS +INTEGRATION survey, implemented by the International Centre for Migration Policy Development (ICMPD, Roland Hosner and Veronika Bilger), the Vienna Institute for International Economic Studies (wiiw, Michael Landesmann and Sebastian Leitner), and the Karl-Franzens-University Graz (Renate Ortlieb).

We thank Gabriele Fischer, Joel Msafiri Francis, and the two anonymous reviewers whose comments helped improve and clarify the manuscript.

## Author Contributions

**Conceptualization:** Daniela Georges, Isabella Buber-Ennser, Judith Kohlenberger, Gabriele Doblhammer.

**Data curation:** Daniela Georges, Isabella Buber-Ennser, Bernhard Rengs.

**Formal analysis:** Daniela Georges, Isabella Buber-Ennser.

**Funding acquisition:** Judith Kohlenberger.

**Investigation:** Bernhard Rengs, Judith Kohlenberger.

**Methodology:** Daniela Georges, Isabella Buber-Ennser, Gabriele Doblhammer.

**Project administration:** Daniela Georges, Bernhard Rengs.

**Resources:** Bernhard Rengs.

**Software:** Daniela Georges, Isabella Buber-Ennser, Bernhard Rengs.

**Supervision:** Bernhard Rengs, Judith Kohlenberger, Gabriele Doblhammer.

**Validation:** Daniela Georges, Isabella Buber-Ennser, Bernhard Rengs.

**Visualization:** Isabella Buber-Ennser, Bernhard Rengs.

**Writing – original draft:** Daniela Georges, Isabella Buber-Ennser, Bernhard Rengs, Judith Kohlenberger, Gabriele Doblhammer.

**Writing – review & editing:** Daniela Georges, Isabella Buber-Ennser, Bernhard Rengs, Judith Kohlenberger, Gabriele Doblhammer.

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
