## [Decision Letter · Decision Letter 0]

3 Jul 2020

PONE-D-20-06839

Health determinants among refugees in Austria and Germany: a propensity-matched comparative study for Syrian, Afghan and Iraqi refugees

PLOS ONE

Dear Dr. Georges,

Thank you for submitting your manuscript to PLOS ONE. After careful consideration, we feel that it has merit but does not fully meet PLOS ONE’s publication criteria as it currently stands. Therefore, we invite you to submit a revised version of the manuscript that addresses the points raised during the review process.

The manuscript covers an important topic, however more details are required towards the item “self-rated health (SRH)”, which is the key variable: At least the differentiation between somatic or psychiatric health is required; significant literature is published on the high prevalence of psychiatric disorders in refugees/asylum seekers originated from the countries you investigate. More precise information on the addressed questions towards “health” needs to be provided in order to be able to interpret the results – eg affective disorder highly influence the self-perception and can bias given results.

 Also information needs to be provided how you counted for the different literacy rates eg in Afghanistan in general, and specifically between men/women – how could you assure that all participants in your target group understood the questions; you covered a certain group of  AS & R regarding educational background, where you need to comment specifically on the significant difference between Germany and Austria regarding low level education, which includes the item “no answer” with 43% in Germany  (table 1). This could also refer to the issue, how the questions were addressed: how many items in your questionnaire; how did you correct for interrater-reliability, considering the different applied languages; how many persons interviewed, what was the time range in the performed interviews – again psychiatric morbidity influences the response rate & content of understanding. Was the instrument tested for validity? 

The literature referral towards # 33,35 is not helpful for the readership –a detailed information on the questionnaire is required in the methods section; also the IRB Nr should be added. Headline & legend to figure 1 is missing.

 In a re-submission the manuscript requires a re-structure: detailed information from tables must not be replicated in the text; the abstract requires a structured scientific presentation; limitations need to emphasized clearly.

We look forward to receiving your revised manuscript.

Kind regards,

Gabriele Fischer, MD

Academic Editor

PLOS ONE

Journal Requirements:

2.We note that you have indicated that data from this study are available upon request. PLOS only allows data to be available upon request if there are legal or ethical restrictions on sharing data publicly. For information on unacceptable data access restrictions, please see http://journals.plos.org/plosone/s/data-availability#loc-unacceptable-data-access-restrictions.

Additional Editor Comments:

The manuscript covers an important topic, however more details are required towards the item “self-rated health (SRH)”, which is the key variable: At least the differentiation between somatic or psychiatric health is required; significant literature is published on the high prevalence of psychiatric disorders in refugees/asylum seekers originated from the countries you investigate. More precise information on the addressed questions towards “health” needs to be provided in order to be able to interpret the results – eg affective disorder highly influence the self-perception and can bias given results.

Also information needs to be provided how you counted for the different literacy rates eg in Afghanistan in general, and specifically between men/women – how could you assure that all participants in your target group understood the questions; you covered a certain group of AS & R regarding educational background, where you need to comment specifically on the significant difference between Germany and Austria regarding low level education, which includes the item “no answer” with 43% in Germany (table 1). This could also refer to the issue, how the questions were addressed: how many items in your questionnaire; how did you correct for interrater-reliability, considering the different applied languages; how many persons interviewed, what was the time range in the performed interviews – again psychiatric morbidity influences the response rate & content of understanding. Was the instrument tested for validity?

The literature referral towards # 33,35 is not helpful for the readership –a detailed information on the questionnaire is required in the methods section; also the IRB Nr should be added. Headline & legend to figure 1 is missing.

In a re-submission the manuscript requires a re-structure: detailed information from tables must not be replicated in the text; the abstract requires a structured scientific presentation; limitations need to emphasized clearly.

Reviewers' comments:

Reviewer's Responses to Questions

**Comments to the Author**

1. Is the manuscript technically sound, and do the data support the conclusions?

Reviewer #1: Yes

Reviewer #2: Yes

2. Has the statistical analysis been performed appropriately and rigorously? 

Reviewer #1: I Don't Know

Reviewer #2: Yes

3. Have the authors made all data underlying the findings in their manuscript fully available?

Reviewer #1: Yes

Reviewer #2: Yes

4. Is the manuscript presented in an intelligible fashion and written in standard English?

Reviewer #1: No

Reviewer #2: Yes

5. Review Comments to the Author

Reviewer #1: This is an interesting and timely manuscript. It has good implications for health policies in Germany. Please see my feedback below:

Pg. 3 line 67 what do you mean by the latter? It is not clear. You may say post-resettlement context or in host country or destination country.

Line 68-69 is hard to follow. Please reword the sentence.

Pg. 3 line 71 spell out GDP

Pg. 4 line 78 now just say GDP in this paragraph.

Pg. 4 line 78 please briefly explain what EU-28 is

Pg. 4 line 93 spell out SRH and determinants of what? You can say determinants of health among AS&R

There should be at least a paragraph explaining your theoretical framework, which I believe is the determinants of health.

Pg.4 line 95 do you mean differences in health outcomes?

Pg. 5 line 99 can you tell a little bit more about IAB-BAMF-SOEP-Refugee Survey and ReHIS ? What kind of questions do they include?

Pg. 5 line 126 please spell out ISCED as well

Pg. 17 line 348, please explain what period effect means.

Pg. 18 line 359 it is unfortunate that even though you have access to these data sets, your analysis was limited to certain variables for the comparison reason.

I think the manuscript would benefit from proofreading, which could greatly improve the flow of the paper.

Reviewer #2: No comments from my side as the plosone online submission system is too complicated to interact in a meaningful way and I have no capacity to take time for it. The paper is interesting and I have a few minor comments but cannot submit them here.

6. PLOS authors have the option to publish the peer review history of their article (what does this mean?). If published, this will include your full peer review and any attached files.

Reviewer #1: No

Reviewer #2: No

---

## [Author Response · Author response to Decision Letter 0]

13 Oct 2020

Manuscript PONE-D-20-06389

Response to Reviewers

Dear Gabriele Fischer,

Thank you for giving us the opportunity to submit a revised draft of the manuscript “Health determinants among refugees in Austria and Germany: a propensity-matched comparative study for Syrian, Afghan, and Iraqi refugees” for publication in PLOS ONE.

We are grateful for the valuable comments we received from you and Reviewer#1. We have incorporated most of the comments made by you and the Reviewer. Those changes are highlighted within the manuscript by tracked changes. Below, in blue, we explain how we have incorporated the comments in the revised version of our paper. All page numbers and line numbers refer to the clean copy of the revised manuscript.

Editor’s and Reviewer’s Comments to the Authors:

(1) Editor Comments

The manuscript covers an important topic, however more details are required towards the item “self-rated health (SRH)”, which is the key variable: At least the differentiation between somatic or psychiatric health is required; significant literature is published on the high prevalence of psychiatric disorders in refugees/asylum seekers originated from the countries you investigate. More precise information on the addressed questions towards “health” needs to be provided in order to be able to interpret the results – eg affective disorder highly influence the self-perception and can bias given results.

Author response: 

Thank you for pointing this out. We have provided the exact wording and the possible answers for SRH in the two surveys (pg. 7, ll. 147-150), and added information regarding the validity of SRH (pg. 23, ll. 455-456). We also discuss the interpretation of SRH (pg. 23, ll. 456-470). A precise differentiation of the explanatory power of SRH is not possible, as our data suggest that both somatic and psychiatric health are highly correlated with SRH. Previous research tends to suggest that among AS&R, mental health problems have a somewhat stronger impact on SRH than physical health.

Also information needs to be provided how you counted for the different literacy rates eg in Afghanistan in general, and specifically between men/women – how could you assure that all participants in your target group understood the questions; you covered a certain group of AS & R regarding educational background, where you need to comment specifically on the significant difference between Germany and Austria regarding low level education, which includes the item “no answer” with 43% in Germany (table 1). 

Author response: 

We agree with the reviewer’s assessment, and have provided additional information about the data collection of the two surveys to clarify the data quality and the sources of misunderstanding (pg. 21, ll. 403-419). Moreover, we have added a figure to illustrate the educational differences in Germany and Austria by nationality and gender, including a differentiation between “no answer” and “low level of education" (S1 Fig). As described in the original manuscript, we have calculated weights adjusting for education (applying these weights only slightly affected the results). Our models were adjusted for nationality, education, and sex; and we integrated “education” as a matching variable into the propensity score matching. Finally, we discussed potential bias by education and measures to minimize educational bias as applied in the IA-BAMF-SOEP and ReHIS (pg. 22, ll. 427-444). 

This could also refer to the issue, how the questions were addressed: how many items in your questionnaire; how did you correct for interrater-reliability, considering the different applied languages; how many persons interviewed, what was the time range in the performed interviews – again psychiatric morbidity influences the response rate & content of understanding. Was the instrument tested for validity? 

Author response:

Thank you for this suggestion. We have provided additional information about the questionnaires (including online links to the questionnaires, pg. 6-7, ll. 123-145) and discussed the impact of the different ways the data were collected (CATI interviews in Austria and CAPI interviews with interviewers in Germany; more questions and more time spent in the interviews in the IAB-BAMF-SOEP Survey than in the ReHIS; differences in the languages and the translation; and possible bias due to morbidity or literacy) (pg. 21, ll. 403-419; pg. 23, ll. 454-455; pg. 23, ll. 460-470). Both the ReHIS and the IAB-BAMF-SOEP Survey were pretested. While higher rates of non-response among individuals with lower levels of education and poor health, as discussed in the literature, are likely, they cannot be reconstructed by us. Similarly, as the languages used interfere with the country of origin (Syrians and Iraqis were interviewed in Arabic, Afghans mainly if Dari-Farsi), we are not able to disentangle possible differences between languages on the one hand and country of origin on the other. We do not classify interrater-reliability as a problem in our analysis and data because the analyzed information are based on the respondents’ self-disclosure.

The literature referral towards # 33,35 is not helpful for the readership –a detailed information on the questionnaire is required in the methods section; 

Author response:

As suggested, we have included detailed information on the questionnaires in the methods section (pg. 5, ll. 96-114) and added additional literature (pg. 5, ll.106 and 114).

also the IRB Nr should be added. 

Author response:

We appreciate this comment, but our study was exempt from IRB review as we only used de-identified data.

Headline & legend to figure 1 is missing.

Author response:

Thank you for pointing this out. The headline and the legend for figure 1 have been included (pg. 15, ll. 306-309).

In a re-submission the manuscript requires a re-structure: detailed information from tables must not be replicated in the text; the abstract requires a structured scientific presentation; limitations need to emphasized clearly.

Author response:

As suggested, we have dropped detailed information from the tables (especially information on confidence intervals) in the results section (pg. 10-13). We have adopted the presentation of the abstract (pg. 2), and we have emphasized the limitations (pg. 20-23).

(2) Reviewer#1 Comments

Reviewer #1: This is an interesting and timely manuscript. It has good implications for health policies in Germany. Please see my feedback below:

Author response:

Thank you.

Pg. 3 line 67 what do you mean by the latter? It is not clear. You may say post-resettlement context or in host country or destination country.

Author response:

Thank you for pointing this out. We have clarified that we mean “destination country” (pg. 3, l. 62).

Line 68-69 is hard to follow. Please reword the sentence.

Author response:

As suggested, we have reformulated this sentence to make it easier to understand (pg. 3, ll. 63-65).

Pg. 3 line 71 spell out GDP

Author response:

As suggested, we have spelled it out (pg. 4, ll. 66-67).

Pg. 4 line 78 now just say GDP in this paragraph.

Author response:

As suggested, we have used the abbreviation in this paragraph (pg. 4, l. 73).

Pg. 4 line 78 please briefly explain what EU-28 is

Author response:

Thank you. We have briefly explained what the EU-28 is (pg. 4, ll. 74-75).

Pg. 4 line 93 spell out SRH and determinants of what? You can say determinants of health among AS&R

Author response:

As suggested, we have spelled out SRH at this point in the text (pg. 4, l. 89), and thereafter use the abbreviation SRH. Moreover, we have clarified that “determinants” refer to determinants of health among AS&R (pg. 4, l. 89).

There should be at least a paragraph explaining your theoretical framework, which I believe is the determinants of health.

Author response:

We think this is an excellent suggestion. We referred to the WHO’s “Social Determinants of Health” framework, and have explained this framework briefly (p. 7, ll.152-157). 

Pg.4 line 95 do you mean differences in health outcomes?

Author response:

Thank you for pointing this out. Yes, we mean differences in health outcomes, and have inserted this term into the manuscript (pg. 4, l. 89).

Pg. 5 line 99 can you tell a little bit more about IAB-BAMF-SOEP-Refugee Survey and ReHIS ? What kind of questions do they include?

Author response:

Thank you for this important suggestion. As we mentioned above, we have provided information on the questions included in the two surveys (pg. 6-7, ll. 123-145).

Pg. 5 line 126 please spell out ISCED as well

Author response:

Thank you. We have spelled out ISCED when first mentioned (p. 7, l. 163).

Pg. 17 line 348, please explain what period effect means.

Author response:

Thank you for pointing this out. We have added a detailed explanation of what “period effect” means (pg. 20-21, ll. 396-402).

Pg. 18 line 359 it is unfortunate that even though you have access to these data sets, your analysis was limited to certain variables for the comparison reason.

Author response:

We agree that this is an important suggestion, but the small set of variables was necessary due to the explorative and comparative character of our study. Subsequent studies could employ additional variables.

I think the manuscript would benefit from proofreading, which could greatly improve the flow of the paper.

Author response:

Thank you for pointing this out. The manuscript has been proofread.

(3) Reviewer#2 Comments

Reviewer #2: No comments from my side as the plosone online submission system is too complicated to interact in a meaningful way and I have no capacity to take time for it. The paper is interesting and I have a few minor comments but cannot submit them here.

Author response:

Thank you. We appreciate the effort you took to read our manuscript, and we regret that it was not possible to submit the minor comments.

---

## [Decision Letter · Decision Letter 1]

7 Dec 2020

PONE-D-20-06839R1

Health determinants among refugees in Austria and Germany: a propensity-matched comparative study for Syrian, Afghan, and Iraqi refugees

PLOS ONE

Dear Dr. Georges,

Thank you for submitting your manuscript to PLOS ONE. After careful consideration, we feel that it has merit but does not fully meet PLOS ONE’s publication criteria as it currently stands. Therefore, we invite you to submit a revised version of the manuscript that addresses the points raised during the review process.

We look forward to receiving your revised manuscript.

Kind regards,

Joel Msafiri Francis, MD, MS, PhD

Academic Editor

PLOS ONE

Reviewers' comments:

Reviewer's Responses to Questions

**Comments to the Author**

1. If the authors have adequately addressed your comments raised in a previous round of review and you feel that this manuscript is now acceptable for publication, you may indicate that here to bypass the “Comments to the Author” section, enter your conflict of interest statement in the “Confidential to Editor” section, and submit your "Accept" recommendation.

Reviewer #1: All comments have been addressed

Reviewer #2: (No Response)

2. Is the manuscript technically sound, and do the data support the conclusions?

Reviewer #1: Yes

Reviewer #2: Partly

3. Has the statistical analysis been performed appropriately and rigorously? 

Reviewer #1: Yes

Reviewer #2: Yes

4. Have the authors made all data underlying the findings in their manuscript fully available?

Reviewer #1: Yes

Reviewer #2: No

5. Is the manuscript presented in an intelligible fashion and written in standard English?

Reviewer #1: Yes

Reviewer #2: Yes

6. Review Comments to the Author

Reviewer #1: The manuscript has been greatly improved and all my comments have been addressed. I recommend this paper for publication.

Reviewer #2: "105 migration, educational, and employment biographies of AS&R, as well as on their reasons for

106 fleeing, the routes they took, and their personality traits and attitudes [34,35]."

Comment: Personality traits are a scientifically clearly defined issue, to be diagnosed/documented only by experts and/or validated standard questionnaires, else results are at best dubious- discuss…

“The authors confirm that some access restrictions apply to the data underlying the findings presented here, therefore these data cannot be shared publicly.”

Comment:The authors mention that data details- that are relevant- are published somewhere else, which in my understanding might be against Plosone requirements- For me it is ok, if they are mostly available for checkup, but please summarise at least shortly..

"For reasons of comparability, the current study is restricted to AS&R who are Syrian, Afghan,

118 or Iraqi nationals aged 18-59 years who immigrated between 2013 and 2016. These three

119 nationalities have made up a large share of the asylum seekers in Europe in recent years,

120 especially in Austria and Germany [1]. Our sample comprises 2,854 respondents in Germany

121 and 374 in Austria." Why the difference ? is this equivalent to the relative differences in the countries subpopulation sizes ? How is the gender balance ? Is it reflecting the one in the population of the refugees or locals ?An at least short Description of the recruiting strategy and % of those who refused should be given."

"A short cognitive test was also administered."

Comment:Which one ? by experts or (medical) lay persons ? This point must be discussed …

I do not understand why the else very precise researchers did use an amateur approach to mental health- there are accepted STANDARD scales and questionnaires in that field---the mental health part might be useless though not central to the study…but mental health in refugees and also experiences of violence and torture that are also not included are crucial questions in refugee health, influence perceived health, and have not been covered properly or at all unfortunately--- Also, we do not know, who was actually “certified” (medical findings) sick which might create a different situation then a healthy population or only self "reports" that might equally reflect distress in general..…

I recommend that the authors mention that issues in the limitations, maybe with the excuse that the focus was on perceived health and mainly on the host country health care system in general, the data with the exception of this limitations are interesting and should be published…

Finally, there are a lot of data and an excellent handbook of UNHCR (by Kirmayer) on health models, and idioms of distress on the background of culture based illness behavior, that are highly relevant for the groups explored and for the cultural differences in self-reported health, but not considered at all… again this should be at least discussed, as it cannot be fixed anymore at this stage….

7. PLOS authors have the option to publish the peer review history of their article (what does this mean?). If published, this will include your full peer review and any attached files.

Reviewer #1: No

Reviewer #2: No

---

## [Author Response · Author response to Decision Letter 1]

10 Feb 2021

Dear Joel Msafiri Francis,

First of all, we appreciate that you agreed to act as Academic Editor as Gabriele Fischer, who was originally assigned to handle our submission, was unavailable for the second round of revisions. 

Thank you for giving us the opportunity to further improve our revised draft of the manuscript “Health determinants among refugees in Austria and Germany: a propensity-matched comparative study for Syrian, Afghan, and Iraqi refugees” for publication in PLOS ONE.

We are grateful for the valuable comments we received from you and Reviewer#2. Reviewer#1 had no further comments and recommended to publish the revised daft. We have incorporated most of the comments made by you and Reviewer#2. Those changes are highlighted within the manuscript by tracked changes. Below, we explain how we have incorporated the comments in the revised version of our paper. All page numbers and line numbers refer to the clean copy of the revised manuscript.

(1) Reviewer#1 Comments on the revised manuscript

Reviewer #1: The manuscript has been greatly improved and all my comments have been addressed. I recommend this paper for publication.

Author response:

We highly appreciate the recommendation of Reviewer#1 to publish our manuscript.

(2) Reviewer#2 Comments on the revised manuscript

Reviewer #2: “105 migration, educational, and employment biographies of AS&R, as well as on their reasons for 106 fleeing, the routes they took, and their personality traits and attitudes [34,35]."

Comment: Personality traits are a scientifically clearly defined issue, to be diagnosed/documented only by experts and/or validated standard questionnaires, else results are at best dubious- discuss…

Author response:

Thank you for this objection. In the description of the data, we refer to the information provided by the data holder (reference #35: Brücker/Rother/Schupp, 2018). Presumably, the validity of the results on personality traits might be discussed (particularly when surveying refugees). Since our analyses do not include personality traits, an in-depth discussion would probably go beyond the scope of this paper. Subsequent studies are very welcome to take up and to discuss this dimension. However, to avoid misunderstandings, we have changed “personality traits” to “personality” (l. 109).

“The authors confirm that some access restrictions apply to the data underlying the findings presented here, therefore these data cannot be shared publicly.”

Comment: The authors mention that data details- that are relevant- are published somewhere else, which in my understanding might be against Plosone requirements- For me it is ok, if they are mostly available for checkup, but please summarise at least shortly.

Author response:

Thank you for pointing this out. All relevant data are within the manuscript; the references relate to further details about the studies that are not directly related to our analyses, results and discussion. Unfortunately, there are legal restrictions on sharing a (de-identified) dataset. The data cannot be shared publicly, but are available upon request. As requested by PLOS ONE, we have provided the respective DOIs (doi:10.5684/soep.iab-bamf-soep-mig.2016; doi:10.11587/7LX1BD) to the Editors. The anonymous IDs of the respondents selected for the current study are provided upon request.

"For reasons of comparability, the current study is restricted to AS&R who are Syrian, Afghan,

118 or Iraqi nationals aged 18-59 years who immigrated between 2013 and 2016. These three

119 nationalities have made up a large share of the asylum seekers in Europe in recent years,

120 especially in Austria and Germany [1]. Our sample comprises 2,854 respondents in Germany

121 and 374 in Austria." Why the difference ? is this equivalent to the relative differences in the countries subpopulation sizes ? How is the gender balance ? Is it reflecting the one in the population of the refugees or locals ?

Author response:

Thanks for that comment. The different case numbers are based on the different sample sizes of the two surveys. We have tried to include information about the maximum number of persons of each survey, resulting in an unbalanced sample with regard to host country. Additionally, the sample is unbalanced with regard to gender (as we have described in line 225 and in Table 1). The gender balance does not reflect the one of refugees or locals. In our analyses, we adjusted for sex to compensate for the imbalance. We have mentioned the problems of imbalance (ll. 435). 

And at least a short description of the recruiting strategy and % of those who refused should be given.

Author response:

Thank you for pointing this out. We have added a short description of the recruiting strategies (the IAB-BAMF-SOEP-Refugee Survey 2016 included a random sample based on the German Central Register of Foreign Nationals; ReHIS is an interim survey of a panel) and the respective % of refusal (ll. 102, and ll. 111).

"A short cognitive test was also administered."

Comment: Which one ? by experts or (medical) lay persons ? This point must be discussed …

Author response:

Thanks for this suggestion. We have added details about the cognitive test (ll. 135). Since our analyses do not include these results, we would like to dispense with a discussion of the implementation and the results of this test.

I do not understand why the else very precise researchers did use an amateur approach to mental health- there are accepted STANDARD scales and questionnaires in that field---the mental health part might be useless though not central to the study…but mental health in refugees and also experiences of violence and torture that are also not included are crucial questions in refugee health, influence perceived health, and have not been covered properly or at all unfortunately--- Also, we do not know, who was actually “certified” (medical findings) sick which might create a different situation then a healthy population or only self "reports" that might equally reflect distress in general..…

I recommend that the authors mention that issues in the limitations, maybe with the excuse that the focus was on perceived health and mainly on the host country health care system in general, the data with the exception of this limitations are interesting and should be published…

Author response:

Thank you for pointing this out. We have added a brief explanation of the limitations concerning the measurement of mental health (ll. 483) and emphasized that our focus was on self-rated health (ll. 471). 

Finally, there are a lot of data and an excellent handbook of UNHCR (by Kirmayer) on health models, and idioms of distress on the background of culture based illness behavior, that are highly relevant for the groups explored and for the cultural differences in self-reported health, but not considered at all… again this should be at least discussed, as it cannot be fixed anymore at this stage….

Author response:

Thanks for this suggestion. Cultural differences in self-rated health are relevant. However, these might be less relevant when comparing two host countries as we have done. We have mentioned this in the discussion (ll. 494) and recommend that subsequent studies include cultural differences.

---

## [Decision Letter · Decision Letter 2]

15 Apr 2021

Health determinants among refugees in Austria and Germany: a propensity-matched comparative study for Syrian, Afghan, and Iraqi refugees

PONE-D-20-06839R2

Dear Dr. Georges,

We’re pleased to inform you that your manuscript has been judged scientifically suitable for publication and will be formally accepted for publication once it meets all outstanding technical requirements.

Kind regards,

Joel Msafiri Francis, MD, MS, PhD

Academic Editor

PLOS ONE

Additional Editor Comments (optional):

Reviewers' comments:

Reviewer's Responses to Questions

**Comments to the Author**

1. If the authors have adequately addressed your comments raised in a previous round of review and you feel that this manuscript is now acceptable for publication, you may indicate that here to bypass the “Comments to the Author” section, enter your conflict of interest statement in the “Confidential to Editor” section, and submit your "Accept" recommendation.

Reviewer #1: (No Response)

Reviewer #2: All comments have been addressed

2. Is the manuscript technically sound, and do the data support the conclusions?

Reviewer #1: Yes

Reviewer #2: Yes

3. Has the statistical analysis been performed appropriately and rigorously? 

Reviewer #1: Yes

Reviewer #2: I Don't Know

4. Have the authors made all data underlying the findings in their manuscript fully available?

Reviewer #1: Yes

Reviewer #2: Yes

5. Is the manuscript presented in an intelligible fashion and written in standard English?

Reviewer #1: Yes

Reviewer #2: Yes

6. Review Comments to the Author

Reviewer #1: (No Response)

Reviewer #2: Yes, the key issues have been resolved. Some issues are inherent in the study design and data set, and cannot be changed, but this does not need to prevent publication.

7. PLOS authors have the option to publish the peer review history of their article (what does this mean?). If published, this will include your full peer review and any attached files.

Reviewer #1: No

Reviewer #2: No

---

## [Editor Report · Acceptance letter]

19 Apr 2021

PONE-D-20-06839R2 

Health determinants among refugees in Austria and Germany: a propensity-matched comparative study for Syrian, Afghan, and Iraqi refugees 

Dear Dr. Georges:

I'm pleased to inform you that your manuscript has been deemed suitable for publication in PLOS ONE. Congratulations! Your manuscript is now with our production department. 

Kind regards, 

on behalf of

Dr. Joel Msafiri Francis 

Academic Editor

PLOS ONE